# The Soft Prefabricated Orthopedic Insole Decreases Plantar Pressure during Uphill Walking with Heavy Load Carriage

**DOI:** 10.3390/bioengineering10030353

**Published:** 2023-03-13

**Authors:** Hsien-Te Peng, Li-Wen Liu, Chiou-Jong Chen, Zong-Rong Chen

**Affiliations:** 1Department of Physical Education, Chinese Culture University, Taipei 11114, Taiwan; pxd@ulive.pccu.edu.tw; 2Institute of Labor, Occupational Safety and Health, Ministry of Labor, New Taipei City 22143, Taiwan; 3Department of Occupational Safety and Health, Chang Jung Christian University, Tainan 711301, Taiwan; 4Department of Athletic Performance, National University of Kaohsiung, Kaohsiung 811, Taiwan

**Keywords:** biomechanics, muscle fatigue, foot orthoses, treadmill walking, backpack load carriage, arch support, heel cup

## Abstract

This study aimed to investigate the effect of varying the hardness of prefabricated orthopedic insoles on plantar pressure and muscle fatigue during uphill walking with a heavy backpack. Fifteen healthy male recreational athletes (age: 20.4 ± 1.0 years, height: 176.9 ± 5.7 cm, weight: 76.5 ± 9.0 kg) wore prefabricated orthopedic insoles with foot arch support; a heel cup with medium (MI), hard (HI), and soft (SI) relative hardnesses; and flat insoles (FI). They performed treadmill walking on uphill gradients with 25 kg backpacks. The plantar pressure and surface electromyographic activity were recorded separately, in 30 s and 6 min uphill treadmill walking trials, respectively. The HI, MI, and SI significantly decreased peak plantar pressure in the lateral heel compared to FI. The MI and SI significantly decreased the peak plantar pressure in the fifth metatarsal compared to FI. The MI significantly reduced the pressure–time integral in the lateral heel compared to FI. The HI significantly increased the peak plantar pressure and pressure–time integral in the toes compared to other insoles, and decreased the contact area in the metatarsal compared to SI. In conclusion, a prefabricated orthopedic insole made of soft material at the fore- and rearfoot, with midfoot arch support and a heel cup, may augment the advantages of plantar pressure distribution during uphill weighted walking.

## 1. Introduction

Mountaineering is a popular recreational activity that requires carrying heavy backpacks during long-term uphill walking [1]. The added external load and uphill gradient can increase the stress on the bones, ligaments, and muscles [2]. Therefore, this type of activity has always been highly strenuous and fatiguing for mountain climbers. It also induces biomechanical changes, including increased-propulsion vertical and anterior ground reaction force [1], increased frontal ankle range of motion [1], increased plantar pressure [3,4], and increased low-limb muscle activity [2]. These biomechanical changes cause a potential risk for metatarsal stress fractures [4], blister development [1,5], ankle sprain [1], and muscle fatigue [2]. These risks of injury suggest that neuromuscular and skeletal systems are unable to accommodate the demands of load carriage during uphill walking.

Injuries to the foot are commonly prevented using prefabricated orthopedic insoles, which are inexpensive and convenient [5,6]. Prefabricated orthopedic insoles are generally designed based on features and characteristics including arch support, heel cups, and cushion properties [7,8]. Two studies have shown a decrease in plantar pressure and the risk of foot injury using these devices while carrying a heavy backpack during level walking [5,9]. Peduzzi de Castro, Abreu, Pinto, Santos, Machado, Vaz, and Vilas-Boas [9] designed a plantar pressure relief insole and found that it was effective in reducing foot plantar pressure in the little toes (second to fifth toe), medial midfoot, lateral midfoot, and forefoot. Melia, Siegkas, Levick, and Apps [5] reported that a soft orthopedic insole attenuated plantar pressure of the lateral and medial forefoot and increased the contact area over the whole foot. However, to date, there appears to be a gap in the literature regarding the effect of prefabricated orthopedic insoles on plantar pressure during uphill walking while carrying a heavy backpack.

A previous study found that the muscle activity of the erector spinae, vastus medialis, soleus, and gastrocnemius increased when carrying a load during uphill walking compared with unloading during level walking [2]. The author suggests that muscles require more effort to overcome external loads and uphill gradients, resulting in muscle overexertion, muscle fatigue, and injury. In addition, muscle fatigue may increase the risk of falls and endanger the lives of mountain climbers. Therefore, resisting muscle fatigue and assisting in improving the biomechanical efficiency of walking are critical issues. Previous studies have found that a custom-made orthopedic insole could decrease muscle fatigue during level walking [10], which is associated with improving the mechanical energy that is stored and returned at each step [11]. However, it is unclear whether prefabricated orthopedic insoles help with resistance muscle fatigue while carrying a load during uphill walking.

Prefabricated orthopedic insoles are made of varying materials which induce different biomechanical effects. The stiff orthopedic insole slightly deforms when bearing weight, which results in less impact attenuation and greater plantar pressure; however, a method to control abnormal foot motion is better, such as overpronation of the foot [12,13,14]. Conversely, the soft orthopedic insole can fit the geometrical shape of the plantar because of the characteristics of the material, which is likely to contribute to attenuating impact and foot pressure [5,12,15]. This hypothesis was confirmed by Melia, Siegkas, Levick, and Apps [5], who found that low-density shoe insoles (soft) were better at decreasing plantar pressure than high-density shoe insoles (hard), as they produced more contact area under the whole foot. However, overdeformation of the insole occurs if the material of the insole is too soft, which negatively influences the impact absorption and plantar pressure attenuation [5]. Therefore, the effect of carrying a heavy backpack during uphill walking while wearing prefabricated orthopedic insoles with different hardness values should be considered, because this issue is critical to developing an appropriate prefabricated orthopedic insole for mountaineering activity.

This study aimed to investigate the effect of varying the hardness of a prefabricated orthopedic insole on plantar pressure and muscle fatigue during uphill walking while carrying a heavily loaded backpack. It was hypothesized that a soft prefabricated orthopedic insole may decrease plantar pressure and muscle fatigue.

## 2. Materials and Methods

### 2.1. Subjects

The institutional review board of the Jianan Psychiatric Center, Ministry of Health and Welfare (IRB No. 21-027) approved all experimental procedures for this study. Fifteen healthy male recreational athletes participated in this study (Table 1). The inclusion criteria were that they had practiced back squats with a loaded barbell (greater than 15 kg) or had participated in heavy-loaded carriage (greater than 15 kg) trail walking. Exclusion criteria were a history of asthma, heart disease, or hypertension; any physical illnesses; an injury to the upper or lower limbs within the experimental period; or an injury 6 months prior to the experiment. A priori sample size calculation was performed using a free online tool, G*Power (www.gpower.hhu.de accessed on 1 April 2021), with a power level of 80% and an α level of 0.05 [16]. The expected effect size was calculated using the means (294.1 and 407.7) and standard deviations (172.0 and 250.0) of the average contact pressure of the hallux during uphill walking [7]. This revealed that a sample size of 13 participants would be sufficient for the analysis. All subjects were required to read and sign an informed consent form prior to the experiment.

### 2.2. Insoles

The prefabricated insoles of flat and thin shoes (FI) were used as a control group. Three commercially available prefabricated insoles with medium, hard, and soft relative hardnesses (MI, HI, and SI, respectively) were selected as the experimental groups in this study (Figure 1). These prefabricated orthopedic insoles had the same features as foot arch support and heel cups. The hardness of the prefabricated orthopedic insole was examined in the forefoot, mid-foot, and rear foot areas using a hardness tester (Figure 1) (Teclock GS-709N Type A, Teclock Co., Nagano, Japan). Both hands held the hardness tester and pushed down vertically at the selective area five times during the hardness testing [7].

### 2.3. Experimental Protocol

Participants visited the laboratory 1 day before the experimental session for familiarization and anthropometric measurements. Anthropometric data for each participant, including height, weight, foot type, foot length, and leg type, were collected by the same tester. Foot type was calculated using the Chippaux and Smirak index (CSI) [17,18,19,20], and leg type (knock knee, bow knee, or normal knee) was examined by subjective judgment [21].

This was a randomized single-blind study. The participants completed the experimental sessions in 1 day. At the start of the experimental session, participants were informed to wear the same brand of socks (Footdisc, Inc., Taipei, Taiwan) and shoes (Maximizer16, Mizuno Inc., Taipei, Taiwan) to prevent sock and shoe characteristics from influencing the results. They performed a warm-up consisting of 5 min of dynamic and static stretches and 5 min of riding a stationary bike at a selected speed. After the warm-up, all participants performed 4 uphill walking trials in 4 prefabricated orthopedic insole conditions while carrying a pack frame of 25 kg [22,23], with 15 min of static rest between trials. All trials were performed on a treadmill (NordicTrack X7i, Sydney, Australia) at the same speed and gradient (2.57 km/h and 24% gradient). Each trial was divided into two parts. The first consisted of collecting plantar pressure data for 30 s; the second of collecting surface electromyography data (EMG) for 6 min.

Plantar pressure data were collected at 100 Hz using the Tekscan system (Wireless Tekscan Device, Boston, MA, USA) and analyzed using the F-Scan software (F-Scan Research 4.5) [9,24,25,26]. The footprint was divided into 13 plantar foot regions to analyze the plantar pressure data. The following variables were analyzed for the 13 plantar foot regions of the dominant leg: peak pressure, pressure–time integral, and contact area. The outcome of these variables was the mean of a complete gait cycle, excluding the first and last step of the 30 s plantar pressure data collection trial. The dominant leg was defined as the leg regularly used to kick a ball [27,28,29,30,31,32].

The EMG was recorded at 2000 Hz using wireless surface electromyography (Trigno, Delsys Inc., Boston, MA, USA) [33,34,35,36,37] from the following muscles of the dominant leg: rectus femoris, tibialis anterior, biceps femoris, and gastrocnemius. Before sensor attachment, the hair on the skin’s surface was shaved using a razor and wiped with alcohol. Four electrodes were affixed using kinesiology tape to the selected muscle areas. Raw EMG data were band-pass filtered from 20 to 450 Hz using a fourth-order zero-lag Butterworth filter. To examine the fatigue effect, the raw EMG data were clipped in the first 1 min as pre-fatigue data and in the last 1 min as post-fatigue data. Fast Fourier Transformation of the raw EMG data was used to calculate the power spectral density (*PSD*). The *PSD* was set as follows: window length (0.125 s); window type (Hanning); window overlap (0.0625 s). The median frequency (*MF*) was calculated using Equation (1). The participant wearing the control insole performed 1 min of uphill walking (no external load), which was used to normalize the *MF* data to be expressed as a percentage [38,39].
(1)$$MF=\int_{0}^{MF}PSD\left(f\right)df=\int_{MF}^{\infty }PSD\left(f\right)df=\frac{1}{2}\int_{0}^{\infty }PSD\left(f\right)df$$

*MF*: median frequency; *PSD*: power spectral density.

### 2.4. Statistics

All statistical analyses were performed using the SPSS version 18.0 software (SPSS Science Inc., Chicago, IL, USA). Descriptive statistics (mean ± SD) are presented for all outcome measurements. To assess differences between the four prefabricated orthopedic insole conditions in plantar pressure and hardness of the insole, one-way repeated measures ANOVA, followed by Bonferroni’s post hoc test (*p* ≤ 0.05), were performed. A two-way ANOVA (insole condition × pre- and post-fatigue) was performed on the MF of the four muscles. A post hoc test was conducted using the Bonferroni test (*p* ≤ 0.05). The effect size (ES) was calculated using partial eta squared (η^2^). The partial eta squared was considered the η^2^ of 0.01 small, 0.06 medium, and above 0.14 large [9,40,41,42,43,44]. Statistical power was calculated using SPSS software.

## 3. Results

### 3.1. Hardness of Insole

The hardness values of the insoles are presented in Table 2. The HI was the stiffest, and the SI the softest insole compared with the rest of the insoles in the forefoot, mid-foot, and rear-foot regions. The MI presented greater hardness in the mid-foot and lower hardness in the forefoot and the rear foot than FI.

### 3.2. Peak Plantar Pressure

The peak plantar pressure data for the subject’s dominant leg are presented in Table 3. The HI presented greater peak plantar pressure in the hallux, second, third, fourth, and fifth toes than the other insoles, and showed greater peak plantar pressure in the mid-foot than the SI. The FI presented greater peak plantar pressure in the fifth metatarsal than the MI and SI, and presented greater peak plantar pressure in the lateral heel than the other insoles.

### 3.3. Pressure–Time Integral

The pressure–time integral data of the subject’s dominant leg are presented in Table 4. The HI presented a greater pressure–time integral in the hallux, third, fourth, and fifth toes than the other insoles, and presented a greater pressure–time integral in the second toe than the MI and SI. The FI presented a greater pressure–time integral in the lateral heel compared to the MI.

### 3.4. Contact Area

The contact area data for the dominant leg of the subject are presented in Table 5. The SI had a greater contact area in the first, second, third, and fourth metatarsals than the HI. The HI had a greater contact area in the hallux than FI.

### 3.5. Median Frequency of EMG

The median frequency of the EMG data of the subject’s dominant leg is presented in Table 6. A significant main effect of pre- and post-fatigue was noted on the tibialis anterior (*p* = 0.003, ES = 0.474, power = 0.910). Post-fatigue presented a lower median frequency of EMG than pre-fatigue in all insoles.

## 4. Discussion

The major findings of the current study were that both MI and SI, with lower hardness values in the fore- and rear-foot regions, significantly reduced the peak plantar pressure in the fifth metatarsal and lateral heel, and MI significantly reduced the pressure–time integral in the lateral heel compared with FI. The HI, with the hardest hardness in the fore-, mid-, and rear-foot regions, significantly increased the peak plantar pressure and pressure–time integral in the toes, but decreased the contact area in the metatarsal compared with the other insoles with lower hardness values. The MI, HI, and SI, which were prefabricated orthopedic insoles with foot arch support and heel cups, significantly decreased the peak plantar pressure in the lateral heel compared with the FI. No difference was found among all insoles in terms of reducing muscle fatigue. Thus, these results partially support our hypothesis that the soft prefabricated orthopedic insole decreases plantar pressure, but cannot reduce muscle fatigue.

Previous studies have indicated that soft orthopedic insoles could contribute to reducing plantar pressure during loading gait and level walking [5]. The results of this study further confirmed that soft, prefabricated orthopedic insoles can reduce plantar pressure, specifically in the fore- and rear-foot regions, during loading gait and uphill walking. Based on the obtained plantar pressure data, Peduzzi de Castro, Abreu, Pinto, Santos, Machado, Vaz, and Vilas-Boas [9] used finite element analysis to develop insoles for obese and weight-bearing individuals, and found that insoles configured with soft cork gel in the forefoot and heel can redistribute the plantar pressure in these areas. For example, the pressure of the rearfoot of the weight-bearing person is transferred from the lateral to the medial heel, and the pressure of the forefoot and the lateral mid-foot of an obese person is decreased. The relatively reduced plantar pressures observed in the toes and forefoot during weight-bearing walking may decrease the risk of foot blisters [25,45], metatarsalgia, and stress fractures in these foot regions [46], as well as plantar fasciitis [47]. Nevertheless, softer is not always better, as the MI significantly reduced the pressure–time integral in the lateral heel, but the SI did not. The MI showed greater hardness in the fore- and midfoot regions than the SI did. This could partly contribute to the stiff arch support of the midfoot MI.

Wearing MI, HI, and SI was shown to decrease peak plantar pressure in the lateral heel compared with FI during weight-bearing uphill walking in this study. The prefabricated orthopedic insoles used herein featured arch support and deep heel cups. Previous research has shown that a flat foot wearing a custom-made arch support insole can reduce the average pressure and peak pressure in the rearfoot by 9.6% and 17%, respectively [48], which is similar to the results of a study that achieved reduced peak pressure in the rear foot. The outcome of the current study was not in accordance with previous studies reporting that orthopedic insoles cannot decrease plantar pressure in the rearfoot during weight-bearing level walking [5,9]. This could be attributed mainly to the different features, such as the presence of arch support and deep or shallow heel cups of the orthopedic insoles, rather than the hardness of the insoles.

This study demonstrated that the SI showed a greater contact area in the metatarsal compared with the HI during weight-bearing uphill walking. Previous research has indicated that insoles with high hardness values are mainly used for foot realignment, while insoles with low hardness values are mainly used for foot shock absorption [49]. Low-density materials can easily reduce peak plantar pressure, and their soft and compliant properties allow the foot to better adapt to the load on the plantar geometry and produce a more uniform pressure distribution [50]. The use of insoles can increase the contact area and reduce the peak pressure on the foot [51]. The SI, with soft material in the forefoot, was considered to fit the geometry of the metatarsal, causing uniform pressure distribution. Moreover, the increased contact area was subjectively considered to increase the comfort of the insoles [5].

This study found that tibialis anterior fatigue significantly increased after weight-bearing uphill walking in all insoles. This result is consistent with those of previous studies [39,52]. Previous studies have indicated that the tibialis anterior can ensure adequate space between the foot and ground during the swing phase; correctly align ankle and calcaneus at initial heel contact; and eccentrically drop the mid- and forefoot to the ground during the stance phase, thus avoiding tumbling and falling during weight-bearing uphill walking [39]. Moreover, there was a trend toward decreased tibialis anterior fatigue in the MI (1.87%) after weight-bearing uphill walking compared with the HI, SI, and FI (2.58%, 4.12%, and 6.99%, respectively). The slight decrease in tibialis anterior fatigue while wearing the MI may imply that there is a decreased demand for absorption of impact forces of the muscles of the hip and knee [39].

The current study had some limitations. This study was a short-term test and was only performed on a treadmill indoors, which may not be sufficiently similar to the outdoor environment. A pack frame with only one loading weight (25 kg) was used, and the results may not be applicable to other weights. Nonetheless, all participants’ perceptions of effort in carrying out the tests using a 25 kg load reached the hard/heavy level on the Borg rating of perceived exertion scale (15.4 ± 0.6). The subjects were healthy male recreational athletes; thus, the results may not apply to other populations.

## 5. Conclusions

The prefabricated orthopedic insoles with lower hardness values demonstrated benefits in the form of decreased plantar pressure at the toes, fifth metatarsal, and lateral heel, as well as increased contact area at the metatarsal during uphill walking while carrying a heavily loaded backpack. However, the prefabricated orthopedic insole was unable to significantly decrease muscle fatigue. It was suggested that the prefabricated orthopedic insole made of soft material at the fore- and rearfoot with midfoot arch support and heel cups may augment the advantages of plantar pressure distribution during uphill walking with a heavy load. These results recommend that mountain climbers wear prefabricated orthopedic insole with lower hardness values during mountaineering when carrying a heavily loaded backpack, as this will decrease the risk of foot injuries, including metatarsal stress fractures, blister development, and sprained ankles.

## Figures and Tables

**Figure 1 bioengineering-10-00353-f001:**
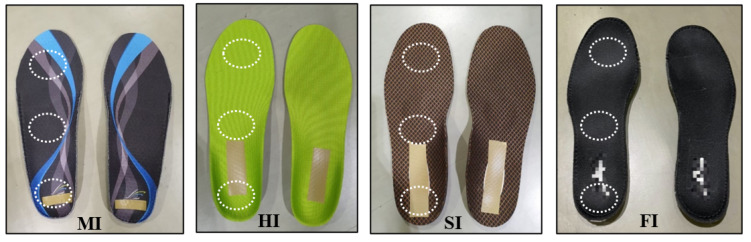
The medium (MI), hard (HI), and soft (SI) relative hardnesses, and the flat insole (FI). (The dotted area indicates the area of measurement for the hardness).

**Table 1 bioengineering-10-00353-t001:** Participants’ characteristics.

Participants (*n* = 15)	
age (years)	20.4 ± 1.0
height (cm)	176.9 ± 5.7
weight (kg)	76.5 ± 9.0
foot length (cm)	26.45 ± 1.28
foot type (*n*)	
high arch	3
low arch	10
normal arch	2
leg type (*n*)	
knock knee	3
bow knee	3
normal knee	9

**Table 2 bioengineering-10-00353-t002:** Hardness of four insoles (HA).

	MI	HI	SI	FI	*p* Value	Effect Size	Power
Fore-foot	25.4 ± 1.14 ^b,c^	52.4 ± 0.54 ^a,c,d^	22.0 ± 0.7 ^a,b,d^	26.8 ± 1.09 ^b,c^	0.001	0.996	1.00
Mid-foot	29.4 ± 1.14 ^b,c,d^	51.4 ± 0.54 ^a,c,d^	16.6 ± 1.81 ^a,b,d^	24.2 ± 1.64 ^a,b,c^	0.001	0.999	1.00
Rear-foot	20.0 ± 0.7 ^b,d^	50.0 ± 0.7 ^a,c,d^	19.6 ± 0.89 ^b,d^	26.6 ± 1.14 ^a,b,c^	0.001	0.997	1.00

^a^ indicates significance compared with MI, ^b^ indicates significance compared with HI, ^c^ indicates significance compared with SI, ^d^ indicates significance compared with FI, *p* < 0.05.

**Table 3 bioengineering-10-00353-t003:** Peak plantar pressure (KPa).

	MI	HI	SI	FI	*p* Value	Effect Size	Power
hallux	154.13 ± 45.18 ^b^	248.66 ± 81.95 ^a,c,d^	159.26 ± 63.60 ^b^	134.53 ± 78.84 ^b^	0.001	0.614	1.00
2nd toe	76.00 ± 21.77 ^b^	131.66 ± 52.55 ^a,c,d^	87.06 ± 39.80 ^b^	83.80 ± 40.75 ^b^	0.006	0.425	0.848
3rd toe	57.40 ± 27.48 ^b^	105.66 ± 32.93 ^a,c,d^	62.80 ± 34.60 ^b^	66.00 ± 62.38 ^b^	0.001	0.480	1.00
4th and 5th toe	49.13 ± 26.93 ^b^	85.86 ± 44.37 ^a,c,d^	55.93 ± 34.98 ^b^	50.33 ± 34.89 ^b^	0.001	0.537	1.00
1st metatarsal	214.26 ± 111.68	231.53 ± 80.46	197.20 ± 89.84	206.73 ± 74.56	0.196	0.104	0.397
2nd metatarsal	169.53 ± 52.18	179.93 ± 47.20	168.60 ± 32.98	173.80 ± 49.24	0.621	0.041	0.163
3rd metatarsal	160.60 ± 49.07	182.40 ± 73.95	154.53 ± 42.07	175.53 ± 65.53	0.082	0.146	0.558
4th metatarsal	116.13 ± 33.69	136.86 ± 56.79	117.93 ± 30.60	130.53 ± 46.43	0.203	0.113	0.238
5th metatarsal	83.06 ± 26.54 ^d^	90.80 ± 27.27	84.26 ± 18.12 ^d^	108.86 ± 36.68 ^a,c^	0.003	0.279	0.913
mid-foot	104.40 ± 47.75	121.40 ± 37.14 ^c^	91.26 ± 18.29 ^b^	111.13 ± 34.74	0.020	0.207	0.759
medial heel	133.73 ± 50.13	128.40 ± 42.99	131.60 ± 34.98	160.53 ± 55.52	0.062	0.159	0.604
lateral heel	104.20 ± 26.72 ^d^	104.86 ± 43.81 ^d^	108.80 ± 35.41 ^d^	140.53 ± 52.71 ^a,b,c^	0.001	0.374	0.988

^a^ indicates significance compared with MI, ^b^ indicates significance compared with HI, ^c^ indicates significance compared with SI, ^d^ indicates significance compared with FI, *p* < 0.05.

**Table 4 bioengineering-10-00353-t004:** Pressure–time integral (Kpa × s).

	MI	HI	SI	FI	*p* Value	Effect Size	Power
hallux	24.10 ± 6.10 ^b^	43.73 ± 11.99 ^a,c,d^	27.00 ± 7.57 ^b^	24.32 ± 10.95 ^b^	0.001	0.617	0.993
2nd toe	14.23 ± 4.80 ^b^	25.08 ± 8.71 ^a,c^	17.20 ± 5.89 ^b^	16.84 ± 7.97	0.001	0.401	0.994
3rd toe	11.14 ± 4.53 ^b^	21.70 ± 7.05 ^a,c,d^	12.26 ± 5.59 ^b^	11.98 ± 10.33 ^b^	0.001	0.515	1.00
4th and 5th toe	8.12 ± 3.80 ^b^	16.31 ± 9.27 ^a,c,d^	10.01 ± 6.97 ^b^	7.68 ± 6.40 ^b^	0.001	0.489	1.00
1st metatarsal	39.98 ± 18.69	46.14 ± 16.50	40.82 ± 15.49	43.15 ± 17.94	0.274	0.087	0.331
2nd metatarsal	33.90 ± 9.33	36.78 ± 8.37	36.42 ± 8.25	38.40 ± 10.91	0.418	0.064	0.254
3rd metatarsal	32.46 ± 7.62	35.76 ± 9.87	33.65 ± 7.68	37.51 ± 11.42	0.193	0.105	0.401
4th metatarsal	26.09 ± 6.00	30.39 ± 7.18	28.42 ± 5.74	31.43 ± 10.77	0.125	0.126	0.483
5th metatarsal	20.38 ± 6.33	25.48 ± 10.67	23.32 ± 6.60	27.76 ± 12.58	0.041	0.177	0.666
mid-foot	26.94 ± 6.58	31.68 ± 8.25	26.00 ± 4.59	29.10 ± 5.78	0.024	0.200	0.739
medial heel	34.02 ± 15.27	32.56 ± 9.30	35.31 ± 10.59	39.23 ± 12.32	0.354	0.074	0.279
lateral heel	25.44 ± 4.98 ^d^	26.35 ± 9.67	28.17 ± 10.25	32.62 ± 10.74 ^a^	0.034	0.184	0.692

^a^ indicates significance compared with MI, ^b^ indicates significance compared with HI, ^c^ indicates significance compared with SI, ^d^ indicates significance compared with FI, *p* < 0.05.

**Table 5 bioengineering-10-00353-t005:** Contact area (cm^2^).

	MI	HI	SI	FI	*p* Value	Effect Size	Power
hallux	6.52 ± 1.88	7.44 ± 1.52 ^d^	5.92 ± 2.44	5.25 ± 2.24 ^b^	0.001	0.348	0.977
2nd toe	3.38 ± 0.89	3.27 ± 0.81	3.60 ± 1.36	3.59 ± 1.07	0.741	0.029	0.126
3rd toe	2.59 ± 0.87	2.89 ± 0.73	2.98 ± 1.14	2.74 ± 1.36	0.482	0.036	0.103
4th and 5th toe	1.53 ± 1.07	2.04 ± 0.72	1.51 ± 0.84	1.60 ± 1.26	0.069	0.154	0.586
1st metatarsal	14.54 ± 1.87	13.73 ± 1.86 ^c^	15.66 ± 2.51 ^b^	14.82 ± 1.67	0.004	0.270	0.899
2nd metatarsal	9.14 ± 1.13	8.54 ± 1.23 ^c^	10.07 ± 1.51 ^b^	9.61 ± 1.56	0.001	0.346	0.976
3rd metatarsal	9.14 ± 1.53	8.66 ± 1.29 ^c^	9.98 ± 1.42 ^b^	9.46 ± 1.76	0.010	0.234	0.830
4th metatarsal	8.19 ± 1.66	7.74 ± 1.88 ^c^	9.08 ± 1.53 ^b^	8.42 ± 1.92	0.048	0.170	0.644
5th metatarsal	7.92 ± 2.84	7.22 ± 2.36	8.75 ± 1.67	8.07 ± 2.83	0.188	0.107	0.406
mid-foot	42.83 ± 10.43	41.85 ± 8.89	44.55± 9.69	41.54 ± 11.03	0.631	0.04	0.16
medial heel	21.17 ± 2.35	21.34 ± 1.75	20.30 ± 1.96	21.80 ± 2.67	0.202	0.103	0.391
lateral heel	21.58 ± 3.46	21.39 ± 2.22	19.93 ± 2.56	21.96 ± 3.24	0.100	0.137	0.523

^a^ indicates significance compared with MI, ^b^ indicates significance compared with HI, ^c^ indicates significance compared with SI, ^d^ indicates significance compared with FI, *p* < 0.05.

**Table 6 bioengineering-10-00353-t006:** Median frequency of EMG (%).

		MI	HI	SI	FI
AT ^e^	T1	94.27 ± 22.13	97.51 ± 12.67	95.39 ± 19.61	98.22 ± 10.75
	T2	92.40 ± 22.87	94.93 ± 12.66	91.27 ± 18.84	91.23 ± 15.51
GA	T1	103.94 ± 16.10	102.39 ± 23.10	102.85 ± 18.49	100.70 ± 7.77
	T2	106.86 ± 14.61	106.44 ± 18.54	96.84 ± 23.10	99.00 ± 14.64
RF	T1	99.83 ± 25.04	103.06 ± 17.67	100.39 ± 10.11	105.78 ± 14.39
	T2	101.12 ± 20.05	101.50 ± 10.49	98.10 ± 12.46	106.13 ± 15.35
BF	T1	91.29 ± 21.45	85.28 ± 21.50	92.12 ± 12.59	90.15 ± 14.71
	T2	93.36 ± 19.59	89.94 ± 16.80	94.66 ± 12.83	87.61 ± 20.05

AT = anterior tibialis, GA = gastrocnemius, RF = rectus femoris, BF = biceps femoris, T1 = pre-fatigue, T2 = post-fatigue, ^e^ indicates significant main effect between T1 and T2, *p* < 0.05.

## Data Availability

The datasets used and/or analyzed during the current study are available from the corresponding authors upon reasonable request.

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
