# Peer review of "The Soft Prefabricated Orthopedic Insole Decreases Plantar Pressure during Uphill Walking with Heavy Load Carriage"

_bioengineering, 2023, doi:10.3390/bioengineering10030353_

Round 1

Reviewer 1 Report

Dear Authors

The study is interesting and highlights the use of insoles to improve the biomechanics of movement in individuals who practice mountaineering. However, I believe that the authors need to make some adjustments to improve the article and clarify some doubts.

Participants were recreationally trained in what modality? Mountaineering? Strength training? What is the weekly frequency? What is the minimum training time (6 months)?

The authors must explain the definition of the 25 kg load in the backpack. The 25 kg load can be light for some and heavy for others. What was the participants' perception of effort in carrying out the tests using a 25 kg load?

Why did the authors not define the load according to the percentage of the total body mass of each participant?

Another important factor for the study is related to the characteristics of the sample participants. The authors report that anthropometric tests were performed, including height, weight, foot type, foot length, and leg type, were collected by the same tester.

Authors need to present these data, as they may influence test results.

Authors should not repeat in the text the values of the variables that are already presented in the tables.

In table 1: the authors must place the unit of measurement of the variable values

In table 3: authors should verify if the symbols of statistical significance were placed correctly (check the variables 2nd toe, 5th metatarsal and mid-foot)

In table 5: remove from the legend “EF = effect size”

In conclusion: the authors must present an applicability of the study.

Sincerely

Author Response

Dear Authors

  1. The study is interesting and highlights the use of insoles to improve the biomechanics of movement in individuals who practice mountaineering. However, I believe that the authors need to make some adjustments to improve the article and clarify some doubts.

A: Thank you for your suggestion.

  1. Participants were recreationally trained in what modality? Mountaineering? Strength training? What is the weekly frequency? What is the minimum training time (6 months)?

A: Thank you for your suggestion. Fifteen participants in this study included ten participants who were experienced at least one-year strength training and five participants who were experienced at least one-year mountaineering. The strength training participant trained at least two days per week and the mountaineering participant climbed mountain at least once a week.

  1. The authors must explain the definition of the 25 kg load in the backpack. The 25 kg load can be light for some and heavy for others. What was the participants' perception of effort in carrying out the tests using a 25 kg load?

A: Thank you for your suggestion. 25kg load in the backpack was used based on a report of Institute of Labor, Occupational Safety and Health, Ministry of Labor in Taiwan. The report indicated that 25kg load is limited for porters to prevent injury of muscle skeleton in Taiwan. Moreover, in the mountaineering regulations of the Islamic Republic of Pakistan, the maximum load for porters is 25 kg. The references have cited in the method. Please refer to line 124. All participants' perception of effort in carrying out the tests using a 25 kg load reached hard/heavy level in Borg rating of perceived exertion scale. It is described in the limitation. Please refer to line 270-272.

  1. Why did the authors not define the load according to the percentage of the total body mass of each participant?

A: Thank you for your suggestion. 25kg load in the backpack was used based on a report of Institute of Labor, Occupational Safety and Health, Ministry of Labor in Taiwan. The report indicated that 25kg load is limited for porters to prevent injury of muscle skeleton in Taiwan. Moreover, in the mountaineering regulations of the Islamic Republic of Pakistan, the maximum load for porters is 25 kg. The references have cited in the method. Please refer to line 124. All participants' perception of effort in carrying out the tests using a 25 kg load reached hard/heavy level in Borg rating of perceived exertion scale. It is described in the limitation. Please refer to line 270-272.

  1. Another important factor for the study is related to the characteristics of the sample participants. The authors report that anthropometric tests were performed, including height, weight, foot type, foot length, and leg type, were collected by the same tester.

A: Thank you for your suggestion. These data are presented in Table 1.

  1. Authors need to present these data, as they may influence test results.

A: Thank you for your suggestion. These data are presented in Table 1.

  1. Authors should not repeat in the text the values of the variables that are already presented in the tables.

A: Thank you for your suggestion. The statistical value of text has been deleted. Please refer to line 166-194.

8.In table 1: the authors must place the unit of measurement of the variable values

A: Thank you for your suggestion. The unit of measurement has been placed in Table 2.

  1. In table 3: authors should verify if the symbols of statistical significance were placed correctly (check the variables 2nd toe, 5th metatarsal and mid-foot)

A: Thank you for your suggestion. We have confirmed the statistical table in SPSS, the pairwise comparisons were no significant for 2nd toe, 5th metatarsal and mid-foot.

  1. In table 5: remove from the legend “EF = effect size”

A: Thank you for your suggestion. The legend “EF has been removed. Please refer to Table 6.

  1. In conclusion: the authors must present an applicability of the study.

A: Thank you for your suggestion. The applicability has written in conclusion. Please refer to line 281-284.

Reviewer 2 Report

This is a well presented and described investigation of the impact of different hardness orthopedic insoles on plantar pressure and muscle function in men walking on an inclined treadmill carrying a weighted pack.

Introduction -provides relevant background, rationale, aim and hypotheses

Materials and methods -

Subjects - was fitness to complete the exercise ascertained to confirm that fatigue was not a large influence in the later tests?

With regard to project design, how was the order of testing of the insoles decided?  Were the different insoles tested in a randomised order for each subject or a standardised order.  If a standard order, what was the order and could this have impacted on the results?

How were the subjects blinded to the insole type?

Is there evidence that 15 mins rest is sufficient between tests?

Results - Participant anthropometric data was collected according to the methods, but these are not reported. Details of the participants should be included to ensure context. A table might be a reasonable presentation of the information.

Overall these are clearly presented and relate to the hypotheses and methods. 

Discussion - This is well organised and discusses the results in the context of the literature.  The limitations section is clear.  

There would be value to discussing the potential difference in loading patterns and fatigue on a treadmill compared to overground walking on an incline, to ensure that the results are evaluated in the light of the differences between treadmill and overground walking, and therefore the context in which the findings can be applied to overground walking.

Conclusion - supported by the results.

Tables and figures - all relevant and clear

Minor detail - 

Page 2, line 20 - It seems that there is a word missing after 'plantar'  Should this be plantar aspect?

Author Response

  1. This is a well presented and described investigation of the impact of different hardness orthopedic insoles on plantar pressure and muscle function in men walking on an inclined treadmill carrying a weighted pack.

A: Thank you for your encouragement.

  1. Introduction-provides relevant background, rationale, aim and hypotheses

A: Thank you for your encouragement.

Materials and methods -

  1. Subjects - was fitness to complete the exercise ascertained to confirm that fatigue was not a large influence in the later tests?

A: Thank you for your suggestion. Fifteen participants in this study included ten participants who were experienced at least one-year strength training and five participants who were experienced at least one-year mountaineering. The strength training participant trained at least two days per week and the mountaineering participant climbed mountain at least once a week. Participants are well regular trained. Moreover, previous research only provided a 2-min rest between each 4-min trial (Walsh et al., 2022). In the current study, the trial was only lasted 6 min. A 15-min rest between trails should not influence later tests.

Walsh, G. S., & Harrison, I. (2022). Gait and neuromuscular dynamics during level and uphill walking carrying military loads. European journal of sport science22(9), 1364–1373. https://doi.org/10.1080/17461391.2021.1953154

  1. With regard to project design, how was the order of testing of the insoles decided?  Were the different insoles tested in a randomised order for each subject or a standardised order.  If a standard order, what was the order and could this have impacted on the results?

The insoles test order was randomized order by creating a random number table in Excel software.

  1. How were the subjects blinded to the insole type?

A: Thank you for your suggestion. The logo of insoles was cover by tape. Please refer to Figure 1.

  1. Is there evidence that 15 mins rest is sufficient between tests?

A: Thank you for your suggestion.

Fifteen participants in this study included ten participants who were experienced at least one-year strength training and five participants who were experienced at least one-year mountaineering. The strength training participant trained at least two days per week and the mountaineering participant climbed mountain at least once a week. Participants are well regular trained. Moreover, previous research only provided a 2-min rest between each 4-min trial (Walsh et al., 2022). In the current study, the trial was only lasted 6 min. A 15-min rest between trails should be sufficient.

Walsh, G. S., & Harrison, I. (2022). Gait and neuromuscular dynamics during level and uphill walking carrying military loads. European journal of sport science22(9), 1364–1373. https://doi.org/10.1080/17461391.2021.1953154

  1. Results - Participant anthropometric data was collected according to the methods, but these are not reported. Details of the participants should be included to ensure context. A table might be a reasonable presentation of the information.

A: Thank you for your suggestion. These data are present at Table 1.

  1. Overall these are clearly presented and relate to the hypotheses and methods. 

A: Thank you for your encouragement.

  1. Discussion - This is well organised and discusses the results in the context of the literature. The limitations section is clear.  

A: Thank you for your encouragement.

  1. There would be value to discussing the potential difference in loading patterns and fatigue on a treadmill compared to overground walking on an incline, to ensure that the results are evaluated in the light of the differences between treadmill and overground walking, and therefore the context in which the findings can be applied to overground walking.

A: Thank you for your encouragement.

Conclusion - supported by the results.

A: Thank you for your encouragement.

  1. Tables and figures- all relevant and clear

A: Thank you for your encouragement.

Minor detail - 

  1. Page 2, line 20 - It seems that there is a word missing after 'plantar'  Should this be plantar aspect?

A: Thank you for your suggestion. We have not found the word missing after ‘plantar’.

Reviewer 3 Report

General comments

This manuscript aims at investigating the effect of varying the hardness of a prefabricated orthopedic insole on plantar pressure and muscle fatigue during uphill walking when carrying a heavily loaded backpack. Authors manage to fulfill sufficiently their aim.

Minor comments

(Abstract and elsewhere throughout MS) Please, do not start sentences with acronyms;

(Table 1) please, add proper measurement unit;

(Tables 2÷4 footnotes) … significant compared with…

Author Response

General comments

  1. This manuscript aims at investigating the effect of varying the hardness of a prefabricated orthopedic insole on plantar pressure and muscle fatigue during uphill walking when carrying a heavily loaded backpack. Authors manage to fulfill sufficiently their aim.

 A: Thank you for your encouragement.

Minor comments

  1. (Abstract and elsewhere throughout MS) Please, do not start sentences with acronyms;

A: Thank you for your suggesting. The issue has been revised. Please refer to text.

  1. (Table 1) please, add proper measurement unit;

A: Thank you for your suggesting. The issue has been revised. Please refer to Table 2.

  1. (Tables 2÷4 footnotes) … significant compared with…

A: Thank you for your suggesting. The issue has been revised. Please refer to Table 2-5.